# Autoimmune Gastritis and Gastric Microbiota

**DOI:** 10.3390/microorganisms8111827

**Published:** 2020-11-19

**Authors:** Laura Conti, Bruno Annibale, Edith Lahner

**Affiliations:** Medical-Surgical Department of Clinical Sciences and Translational Medicine, Sant’Andrea Hospital, Sapienza University of Rome, via di Grottarossa 1035, 00189 Rome, Italy; lau88.conti@gmail.com (L.C.); bruno.annibale@uniroma1.it (B.A.)

**Keywords:** autoimmune gastritis, atrophic gastritis, gastric microbiota

## Abstract

Autoimmune atrophic gastritis is an organ-specific immune-mediated condition characterized by atrophy of the oxyntic mucosa. Autoimmune atrophic gastritis (AIG) is characterized by a progressive loss of acid-secreting parietal cells leading to hypo-achlorhydria. Due to this peculiar intra-gastric environment, gastric microbiota composition in individuals with autoimmune atrophic gastritis was first supposed and then recently reported to be different from subjects with a normal acidic healthy stomach. Recent data confirm the prominent role of *Helicobacter pylori* as the main bacterium responsible for gastric disease and long-term complications. However, other bacteria than *Helicobacter pylori*, for example, Streptococci, were found in subjects who developed gastric cancer and in subjects at risk of this fearful complication, as well as those with autoimmune gastritis. Gastric microbiota composition is challenging to study due to the acidic gastric environment, the difficulty of obtaining representative samples of the entire gastric microbiota, and the possible contamination by oral or throat microorganisms, which can potentially lead to the distortion of the original gastric microbial composition, but innovative molecular approaches based on the analysis of the hyper-variable region of the 16S rRNA gene have been developed, permitting us to obtain an overall microbial composition view of the RNA gene that is present only in prokaryotic cells.

## 1. Introduction

Autoimmune atrophic gastritis (AIG) is a relatively frequent and often undiagnosed disorder with important and potentially life-threatening consequences from a clinical point of view, ranging from micronutrient deficiencies and severe anemia to such neoplastic complications as gastric cancer and gastric type 1 neuroendocrine tumors. Due to its peculiar intra-gastric environment, characterized by severely impaired gastric acid secretion as a result of gastric oxyntic mucosa atrophy, the gastric microbiota composition in individuals with AIG was first supposed and then recently reported to be different from subjects with a normal acidic stomach, possibly assuming a key role in the development of neoplastic complications. This evidence adds new pieces to a constantly developing puzzle on the knowledge of autoimmune atrophic gastritis, a condition far from being completely investigated, and opens the door to new and intriguing perspectives on the management and possible treatment options of this important condition, which reduces the quality of life of millions of persons all over the world. This review addresses different aspects of AIG, focusing particularly on epidemiology, the clinical picture and management, the relationship between hypochlorhydria and long-term complications, and the interplay between the gastric microbiota, autoimmune gastritis and its clinical consequences, as well as the complicated and still debated role of *Helicobacter pylori* infection, providing an updated summary of recent scientific evidence on this intriguing topic.

## 2. Epidemiology and Clinical Manifestations of Autoimmune Gastritis 

AIG is an organ-specific immune-mediated condition that affects the corpus and fundus of the stomach; AIG is characterized by atrophy of the oxyntic mucosa with subsequent hypochlorhydria, hypergastrinemia, and deficiency of intrinsic factors, leading, in late stages, to pernicious anemia [1]. AIG’s classical histopathological alterations consist of corporal-limited *Helicobacter pylori* (Hp)-negative atrophic gastritis with a spared antrum; sometimes, active Hp infection and/or involved antral mucosa may be observed in AIG, showing overlapping features with the multifocal atrophic gastritis mainly linked to Hp [2]. AIG is a condition that may involve any age group, but more frequently affects the elderly and females; most cases are reported in subjects of Northern European descent [3]. The absence of proactive case-finding strategies for AIG diagnosis, the lack of epidemiological studies, and the frequent indolent disease course, possibly leading to an underestimation of the disease, may contribute to the lack of knowledge of the true prevalence of AIG [4]. Moreover, in most papers published in the recent past, AIG was diagnosed only based on serological biomarkers such as anti-parietal cell or anti-intrinsic factor antibodies (PCA/IFA), pepsinogen, and/or gastrin-17 levels [5,6] without any histological confirmatory diagnosis. Finally, for several years, the diagnosis of AIG was frequently underestimated and mistakenly considered only in cases of pernicious anemia (PA), with macrocytic anemia due to vitamin B_12_ malabsorption usually manifesting itself in the late stage of the disease [7,8]. Based on this scenario, AIG prevalence has been estimated to be ~0.5–4.5% globally, varying widely owing to different diagnostic criteria, ethnical and demographical settings [4].

From a pathological point of view, AIG is thought to be the result of a complex interaction between environmental factors and host-related factors such as genetic susceptibility, but data are scant. An Italian study showed a significantly higher prevalence of HLA-DRB1*03 and HLA-DRB1*04 alleles in patients with AIG than in a healthy control group [9]. By contrast, a Finnish study found an association between AIG and HLA-DRB1*04/HLA-DQB1*03, but not with HLA-DRB1*03 [10]. These HLA haplotypes are also frequently associated with other autoimmune diseases, in particular autoimmune thyroid disease, thereby underlining a common HLA-dependent autoimmune pathway [4].

Despite the advancements in knowledge that have been made in the field of AIG, the trigger precipitating the autoimmune response has not been clarified. The resulting immunological dysregulations involve sensitized CD4+T lymphocytes and PCA/IFA, while gastric corpus/fundus tissue damage results from an antibody-mediated destruction of the parietal cells due to selective targeting of the H+/K+ ATPase proton pump [11]. PCA are of immunoglobulin G type, they are directed against the parietal cell H+/K+ ATPase, and they are mainly considered serological markers of autoimmune gastritis. PCA/IFA positivity is considered a helpful tool for AIG diagnosis. However, detection of those antibodies is not sufficient for AIG diagnosis, because they are not specific and are also found in healthy individuals for in escaped negative thymic selection or in patients with other autoimmune diseases such as type 1 diabetes or thyroid diseases, whereby the AIG prevalence is comparatively three- to fivefold higher [12].

Furthermore, serology against H. pylori (IgG AbHp) may be positive in AIG patients with previous contact with the bacterium or in those previously treated for the infection. When a positive serological titer of AbHp is found in a patient with AIG together with a polymorphonucleate inflammatory infiltrate in the gastric mucosa, an active H. pylori infection should be suspected [13].

From a clinical point of view, AIG has been traditionally considered a silent condition, often suspected due to its hematologic findings, and rarely by the presence of gastrointestinal symptoms. Despite the fact that most patients are pauci- or asymptomatic, several studies have shown that dyspeptic symptoms such as postprandial fullness, early satiety, and nausea are among the most common symptoms complained about by AIG patients [14,15,16]. Most commonly, AIG may be suspected in the presence of an iron deficiency and, in particular, anemia due to iron malabsorption consequent to reduced gastric acid secretion (25–50% of patients with AIG) or, rarely, in the presence of pernicious anemia, which is found in up to 15–25% of AIG patients [17,18,19,20]. Less frequently, AIG patients may complain of neurological symptoms such as paresthesia, abnormal proprioception, numbness, ataxia, cognitive impairment, mood disorders, and psychosis. Neurological symptoms are consequences of vitamin B_12_ deficiency, due to an impairment of sensory and peripheral nerve function linked to a reduced production of succinyl coenzyme A, which is essential for myelin sheath structure [21,22]. Finally, concomitant autoimmune diseases, especially Hashimoto thyroiditis or a positive family history for AIG may contribute to increasing the suspicion of an AIG diagnosis.

## 3. Association with Other Autoimmune Diseases

AIG may be associated with a wide spectrum of autoimmune disorders [4]. Hashimoto thyroiditis (HT) is the most frequent autoimmune disease associated with AIG with a 3–8-fold higher reported prevalence than in the general population [23]. Conversely, the prevalence of AIG is 3–5 times greater in patients with autoimmune thyroid disorders [3,5,6]. The association between gastric and thyroid disorders has been observed since the early 1960s when the frequent co-presence of the thyroid and gastric autoantibodies (anti-thyroperoxidase, anti-thyroglobulin, PCA, IFA) in patients with thyroid disorders and PA led Doniach B and Irvine WJ et al. to coin the expression “thyrogastric syndrome” [24,25,26,27]. The impairment of thyroid follicular cells and gastric parietal cells typical of HT and AIG are, respectively, due to a multifactorial etiology resulting from the association between genetic susceptibility and several environmental factors [28,29]. The specific mechanism leading to thyrocytes and/or parietal cell damage is still poorly understood, but this similar phenomenon can be partly elucidated by the common embryological origin of gastric mucosal and thyroid follicular cells, both developing from the endoderm and sharing some functional and morphological similarities [30,31]. Thyro-gastric autoimmunity is currently considered part of type III polyglandular autoimmune syndromes, which include several endocrine and nonendocrine autoimmune disorders as a consequence of immune tolerance loss [27,32,33]. Type I diabetes mellitus (T1DM) is the second most frequently autoimmune disorder associated with AIG. This association has been confirmed by several studies which, found PCA positivity in 10–15% of children (<18 years of age) and in 15–25% of adult T1DM patients [34,35]. Another study showed that 57% of PCA-positive patients and 10% of PCA-negative patients, out of 88 patients with T1DM (of whom 47 were PCA- positive) undergoing gastroscopy with biopsies, received a histological diagnosis of AIG [36]. Apart from HT and T1DM, the other less commonly associated autoimmune disorders are vitiligo and Addison’s disease, followed by only sparse and scant data mainly resulting from case reports on the association between AIG and rheumatoid arthritis [37], celiac disease [38], and many other autoimmune disorders [39,40].

## 4. The Role of Hypochlorhydria and Long-Term Complications

Gastric acidity has several primary functions as a bactericidal defensive barrier, including digestive and absorptive properties. The progressive destruction of hydrochloric acid-secreting parietal cells is typical of AIG and may lead to a gradual hypochlorhydric state [1]. Hypochlorhydria may result in dietary iron malabsorption [41]. The pH increase due to the weakening of the gastric acid defensive barrier may also result in consequent gastric microbiota composition alterations with potential overgrowth of other bacteria than Hp [42]. Furthermore, AIG is considered a precancerous condition with an increased neoplasm risk, and is also linked to possible intra-gastric changes such as hypochlorhydria and increased oxidative stress as a consequence of persistent inflammatory infiltration of the corpus–fundus of the stomach [43]. The hypochlorhydric state may induce enterochromaffin-like (ECL) cell hyperplasia with a major risk of developing type 1 gastric neuroendocrine tumors over time, at percentages varying from 0.4% to 7%, and gastric adenocarcinoma, with an incidence ranging between 0% and 1.8% per year [44,45,46]. The crucial role of parietal cell secretion in maintaining an acidic intragastric milieu is strictly regulated by both endocrinal and neuronal stimulation via the vagus nerve. Hydrochloric acid secretion is stimulated by gastrin, secreted by gastrin-producing cells in the antrum, and by histamine, produced by ECL cells in the corpus or fundus glands. In the presence of oxyntic mucosa atrophy, hypochlorhydria leads to persistently increased levels of gastrin, a well-known risk factor for ECL cell hyperplasia, dysplasia, and type 1 gastric neuroendocrine tumors [20,47]. While the higher risk of gastric cancer in corpus atrophic gastritis is well defined and linked to Hp, considered the first trigger of a multistep carcinogenic process explained by the Correa Cascade [48], the potential carcinogenic mechanisms associated with AIG are still under debate. In fact, it has been hypothesized that inflammation, dysregulation of the host immune system, and an increase in nitrate and nitrose-producing bacteria, leading to a non-acidic intragastric milieu, as occurs in AIG, may play a role in gastric carcinogenesis [49,50,51]. Nowadays, most of the available evidence about the gastric cancer risk associated with AIG is derived from cohort and case–control studies conducted on patients affected by PA [52,53,54,55,56,57]. According to the recently published European guidelines on the management of epithelial precancerous conditions and lesions, AIG is considered a precancerous condition, and patients should be stratified according to risk groups (OLGA/OLGIM system, family history of gastric cancer, or presence of incomplete intestinal metaplasia) and monitored by gastroscopy with biopsies, according to the updated Sydney system protocol [8], at an interval of 3–5 years [58]. Lastly, the most frequent long-term complication associated with the hypochlorhydric state is the onset of hematological alterations due to iron or vitamin B_12_ malabsorption. Iron absorption is strictly dependent on normal gastric hydrochloric acid secretion, which is essential for the reduction of ferric dietary iron to ferrous iron, a major absorbable iron form [59,60]. Patients with iron-deficiency anemia may be managed by oral iron supplementation and it has been suggested to switch to intravenous iron delivery and blood transfusion only in case of severe anemia or exceptional situations [4,61,62]. In the case of vitamin B_12_ deficiency, intramuscular administration is recommended to obtain an ideal vitamin B_12_ normalization, particularly in patients who complained of neurological symptoms, which are also not always reversible. For maintenance therapy, a Cochrane review did not find any superiority of oral or intramuscular vitamin B_12_ treatment in normalizing serum vitamin B_12_ levels, showing cost-effectiveness in favor of oral treatment, but the trials reviewed included patients irrespective of the cause of vitamin B_12_ deficiency and therefore also patients without AIG or PA [63]. 

## 5. Gastric Microbiota: Historical Aspects and *Helicobacter pylori*

A growing body of literature on gastric microbiota composition has been recently published, but data in this field are still scarce and conflicting. Nowadays, no specific gastric microbiota profiles related to different gastric conditions such as Hp gastritis, chronic atrophic gastritis, autoimmune gastritis, or gastric cancer have been well defined [64]. The stomach was historically considered a sterile organ and an unfavorable bacterial growth environment owing to its very low pH and the proteolytic activity of gastric juice, as well as the antimicrobial function of nitric oxide, produced by salivary nitrate [65]. The discovery of Hp about 40 years ago was the first step towards a paradigm shift [66]. Hp infection was recognized to be the major cause of chronic atrophic gastritis, becoming the most thoroughly investigated component of the gastric microbiota [67], and it was classified as a class I carcinogen by the World Health Organization [68] for its contribution to gastric carcinogenesis, as supposed by Correa Cascade [48]. After Hp discovery, the growing interest in gastric histology and microbiology increased over time and many of the older observations, such as the effect of reduced acid secretion on promoting a diverse gastric flora, began to be investigated [69]. A growing number of culture-dependent and molecular method-based studies comparing different microbial compositions in Hp-positive or -negative subjects, in chronic atrophic gastritis and in stomachs with gastric cancer, were published, aiming to assess gastric microbiota diversity and its possible role in gastric carcinogenesis [70].

## 6. The Role of Innovative Methods for the Detection of the Gastric Microbiota

The interest in the field of gastric microbiota has been widely increasing in recent years. The knowledge of the gastric microbiota and its role in human health and diseases, although still limited, is expanding more and more thanks to the development of molecular-based methods [65]. Gastric microbiota composition is challenging to study, owing to the acidic gastric environment, the difficulty in obtaining a representative sample of the entire gastric microbiota by gastric biopsies or gastric juice samples, and their possible contamination by oral or throat microorganisms, which can potentially lead to the distortion of the microbial composition [67]. In 1946, Barber and Franklin published their efforts to culture bacteria from gastric mucosal swabs for the first time in history [71]. This study began the culture-dependent era of gastric microbiota definition. Unfortunately, the culture-based approach has several limitations and it is influenced by many factors comprising the type of gastric sample that is most often limited to luminal contents rather than mucosa-associated organisms, the different gastric acidic state of the stomach the time of the culture, and the methods used for culture [72]. However, the major limitation of the culture-based approach that the vast majority (~80%) of microbial species are not cultivable [73]. These limits and, in particular, the need to exceed the limit of non-cultivable microorganisms led to the development of new molecular techniques based first on DNA genome sequencing and then on next-generation sequencing and molecular analysis of microbiota [74]. Furthermore, the main advantage of culture-independent methods is the selective detection of viable bacteria. In the era of culture-independent methods, molecular approaches allowed researchers to markedly enhance the study of the gastrointestinal tract in general and the gastric microbiota profile in detail using DNA-based approaches, either relying on whole-genome information or focusing on the 16S rRNA gene as a standard phylogenetic marker [67,75]. Methods relying on the sequence-specific separation of equal-sized 16S rDNA PCR-amplified fragments such as denaturing gradient gel electrophoresis (DGGE), temporal gradient gel electrophoresis (TTGE), or terminal restriction fragment length polymorphism (TRFLP) of polymerase chain reaction (PCR)-amplified 16S rDNA fragments were introduced in 2000 [72,76]. However, the above-cited methods were not able to define sequence differences at the species level and often relied on short 16S rDNA-amplified fragments. Due to these limitations, they were successively replaced in the mid-2000s by a next-generation sequencing technique based on high-throughput DNA sequencing techniques, allowing researchers to assess the bacterial composition rapidly in many samples with greater sequencing depth and sequence coverage. The first study using high-throughput DNA sequencing techniques to analyze the human gastric mucosa microbiota was published in 2008, reporting a higher gastric microbiota diversity in Hp-negative compared to Hp-positive patients [77]. Since then, many other studies showed that Hp eradication may lead to increased bacterial diversity and restore the relative abundance of other bacteria similar to Hp-negative subjects, suggesting that Hp colonization results in alterations to the gastric microbiota, which are reversible by antibiotic treatment [78]. More recently, many innovative molecular approaches based on analyses of the hyper-variable region of the 16S rRNA gene by 16S rRNA gene-restricting high-throughput sequencing methods (Illumina, and Ion Torrent) have been developed, permitting us to obtain an overall microbial composition view of RNA that is only present in prokaryotic cells [74]. The gastric microbiota analysis based on 16S rRNA gene-restricting high-throughput sequencing methods has permitted us to identify many unexpected or previously unknown bacteria in Hp-negative stomachs with 262 phylotypes representing 13 phyla [65,79]. This technology allows us to analyze microbiota composition below the genus level, but only provides information on the bacterial presence, without any detail about bacterial functions or the vitality state of microorganisms [80]. Based on this scenario, the future goal is to perform studies assessing the metabolically active bacteria of the stomach using many innovative and different methods such as reverse-transcribed 16S rRNA as an amplification template [81].

## 7. Gastric Microbiota, Hypochlorhydria, and Autoimmune Gastritis 

In recent years, several studies have been published in the field of gastric microbiota aiming to discover whether substantial differences between a healthy stomach and pathological gastric conditions were found. For this purpose, most of the authors working in this field have conducted studies mainly on the gastric microbiota of patients with gastric cancer, a well-known long-term complication of Hp-related atrophic gastritis, and AIG. Concerning gastric precancerous conditions, gastric microbiota composition of Hp-related atrophic gastritis has been better defined than that of AIG. Hp-induced atrophic gastritis has been reported to display a lower bacterial diversity and a decreased abundance of other microbial groups than a healthy stomach with a high prevalence of Proteobacteria (as Hp itself is a member of this phylum), as shown in previous studies [82,83]. However, this seems to be a reversible situation that may substantially change after Hp eradication. Apart from Hp, a well-established carcinogen linked to gastric cancer, the composition of the gastric microbiota has not yet been investigated thoroughly and conflicting data have emerged from different studies. Moreover, it should be noted that most of the data derive from Asian populations and only rarely from Western populations, and that different sources of gastric samples, different microbial composition-analyzing methods, as well as different reference groups have been used, such as chronic gastritis instead of a healthy stomach, in many studies [65,84,85,86]. A recently published systematic review on the gastric microbiota showed highly heterogeneous results for gastric microbial composition, with 266 bacterial genera identified, of which 57 were more frequently reported in the normal acidic stomach and distributed among five bacterial phyla, including Proteobacteria, Firmicutes, Bacteroidetes, Actinobacteria, Fusobacteria, and the most abundant genera: Helicobacter, Streptococcus, and Prevotella [65].

Considering gastric conditions characterized by a hypochlorhydric state and, in particular, corpus-restricted AIG, where the gastric mucosal barrier becomes progressively compromised due to an immune-mediated pathological mechanism, it has been hypothesized that the progressive loss of the acid barrier function may favor a bacterial overgrowth, thus potentially affecting the gastric microbiota composition; however, data in this field are currently scarce and conflicting (Figure 1) [42,65]. A previous study published by Parsons et al. focused on assessing the diversity of gastric microbial profiles in different hypochlorhydric states, including Hp-induced atrophic gastritis and AIG. This study showed that patients with AIG presented a relatively higher microbial diversity and bacterial abundance than normal stomachs with the largest proportion of Streptococci among the groups investigated [42]. Research on gastric microbiota in AIG is therefore at an early stage. As it has recently been accepted as a precancerous condition to be subjected to endoscopic surveillance for its potential neoplastic complications, further studies are needed to understand if changes in the gastric microbiota could be associated with the progression of gastric carcinogenesis [72].

Regarding the gastric microbiota in subjects with gastric cancer, most studies have shown that it seems to be characterized by an enrichment of bacterial diversity due to the additional colonization of the gastric environment by oral taxa [84,87] such as Streptococcus, Staphylococcus, Lactococcus, Bacillus, Prevotella, Veillonella, and Leptotrichia, as well as intestinal taxa such as Lactobacillus [88], Clostridium and Fusobacterium, with a contemporary decreased presence of Hp [84,85,89]. Interestingly, in the above-cited study on AIG, in individuals with this condition, who are predisposed to gastric cancer, the largest proportion of Streptococci was found; these microbes notably belong to the oral microbiota and, probably, the non-acidic gastric environment due to the fact that hypochlorhydria offers an acceptable habitat, making this oral commensal intriguing because plays a potential role in gastric carcinogenesis. However, conflicting data have been found in Portuguese studies, in which a decrease in Streptococcus in individuals with gastric cancer was reported [70,85,90].

## 8. Conclusions Remarks and Research Agenda

As in other body districts, in the stomach, growing knowledge of the possible role of the microbiota in health and disease is emerging from recent studies. Recent data confirm the prominent role of Hp as the main bacterium responsible for gastric disease and long-term complications. However, other bacteria, and possibly other poorly or not yet investigated viral or fungal microbiota components, are emerging and likely play a role in conditions with an altered intragastric environment such as AIG, which is notably characterized by a non-acidic stomach favoring the overgrowth of microorganisms that are otherwise not viable in the acidic stomach. Some of these bacteria, for example, Streptococci, are found in subjects who have developed gastric cancer and in subjects at risk of this fearful complication, such as those with AIG. These first pieces of evidence certainly cannot be interpreted as a point of arrival, but should rather be viewed as a starting point for future research in this very complex and intriguing field in which much work is yet to be done. In the last few years, many pieces have been added to the knowledge puzzle, and future research is needed. As detailed in Table 1, several aspects are still awaiting clarification, to ultimately pave the way for possible innovative treatment strategies to eventually prevent the progression of AIG or neoplastic complications by therapeutic gastric microbiota modulation.

## Figures and Tables

**Figure 1 microorganisms-08-01827-f001:**
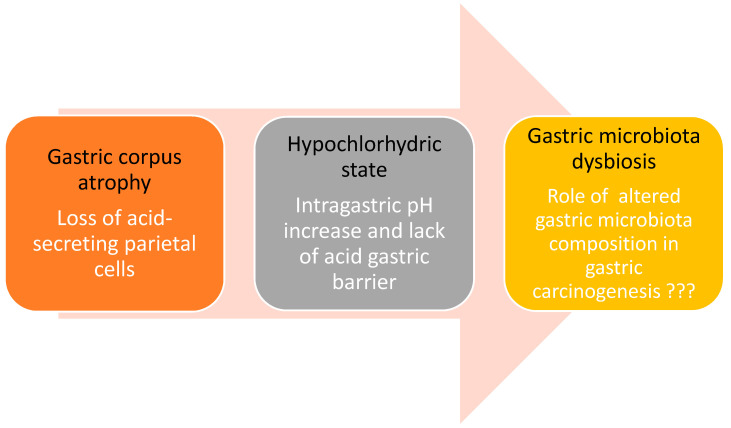
A possible link between autoimmune gastritis, hypochlorhydria, and gastric cancer may be hypothesized. Autoimmune gastritis, a corpus-restricted Hp-negative atrophic gastritis, is characterized by progressive immune-mediated atrophy of the oxyntic mucosa with subsequent loss of acid-secreting parietal cells, leading to hypochlorhydria. This changed intra-gastric microenvironment can make possible the survival of other bacteria than Hp possibly playing a key role in gastric carcinogenesis.

**Table 1 microorganisms-08-01827-t001:** Proposal of research agenda on gastric microbiota.

**1** To standardize techniques and gastric samples used to assess the viable microbiota in the stomach by giving priority to innovative methods based on RNA for sequencing
**2** To perform studies considering possible confounding factors on the gastric microbiota such as drugs and dietary, smoking, and alcohol habits
**3** To perform longitudinal, multicentre studies to increase the knowledge on the role of gastric microbiota in gastric carcinogenesis
**4** To launch studies on the gastric microbiota in Caucasian populations as the available data on Asian populations may not be necessarily comparable and valid in non-Asian subjects

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
