# Peer review of "Autoimmune Gastritis and Gastric Microbiota"

_microorganisms, 2020, doi:10.3390/microorganisms8111827_

Round 1

Reviewer 1 Report

This review focuses on the relationship between autoimmune gastritis and the gastric microbiome. While it presents a good summary of the gaps in knowledge in this emerging field, it perhaps lacks some details on the nature of different organisms in the different disease stages. Perhaps this could be easily rectified with a Table including some of this information? The use of English needs some attention, as some sentences are missing words or do not make sense

Author Response

Point 1: This review focuses on the relationship between autoimmune gastritis and the gastric microbiome. While it presents a good summary of the gaps in knowledge in this emerging field, it perhaps lacks some details on the nature of different organisms in the different disease stages. Perhaps this could be easily rectified with a Table including some of this information? The use of English needs some attention, as some sentences are missing words or do not make sense

Response 1:

 We thank the Reviewer for the comments and the time dedicated to evaluate the review.

Unfortunately, the number of papers on AIG gastric microbiota are too scarce and conflicting to make a specific table which, we fear, might be misleading for reader conferring the wrong impression that the knowledge in this field is established yet.

Nowadays, most of knowledge in this field has focused on the gastric microbiota in gastric cancer and Correa cascade starting with chronic gastritis in Hp colonised individuals which can further progress towards atrophy, intestinal metaplasia, dysplasia and eventually gastric adenocarcinoma.

AIG is thought to be a different entity from Hp atrophic gastritis; AIG and its classical histopathological alterations consist precisely in corpus-limited Helicobacter pylori (Hp)-negative atrophic gastritis and it might be characterized by different carcinogenesis steps. Therefore, the lack of details on the nature of different microorganisms in the different disease stages of AIG is due to the lack of data regarding the gastric microbiota in AIG.

The English revision is in blue

Reviewer 2 Report

The review by Conti et al discusses autoimmune atrophic gastritis (AIG) also in the context of the microbiota present in this condition and in gastric cancer, a condition tha can follow AIG.

The review is well written. I would split in two some of the phrases, also in the abstract, which are too long.

I suggest that the authors stress more the significance of the autoantibodies in the disease and their possible role. It will be intersting to know whether in the literature there is association with HLA Class II or Class I alleles, given that this is an autoimmune condition. Are there any antibodies also against the bacteria? there is anything reported in the literature in this regards?

Author Response

Point 2: The review by Conti et al discusses autoimmune atrophic gastritis (AIG) also in the context of the microbiota present in this condition and in gastric cancer, a condition that can follow AIG. The review is well written. I would split in two some of the phrases, also in the abstract, which are too long. I suggest that the authors stress more the significance of the autoantibodies in the disease and their possible role. It will be interesting to know whether in the literature there is association with HLA Class II or Class I alleles, given that this is an autoimmune condition. Are there any antibodies also against the bacteria? there is anything reported in the literature in this regards?

Response 2:

 We thank the Reviewer for the comments and the time dedicated to evaluate the review. We fully agree with the suggestions. Now, new paragraphs regarding genetic predisposition and the role of autoantibodies in AIG have been included (see page 2, from line 66 to 87); as regards antibodies against gastric bacteria, to our best knowledge, there are no data, except for antibodies against H.pylori (IgG Hp-Ab).

The English revision is in blue